# Anne O'Tate: Value-added PubMed search engine for analysis and text mining

**Neil R. Smalheiser**[1]*, **Dean P. Fragnito**[2], **Eric E. Tirk**[2]

**1** Department of Psychiatry, University of Illinois at Chicago, Chicago, Illinois, United States of America,
**2** Xornet Inc., Rochester, New York, United States of America

* neils@uic.edu

## Abstract

Over a decade ago, we introduced Anne O'Tate, a free, public web-based tool http://arrowsmith.psych.uic.edu/cgi-bin/arrowsmith_uic/AnneOTate.cgi to support user-driven summarization, drill-down and mining of search results from PubMed, the leading search engine for biomedical literature. A set of hotlinked buttons allows the user to sort and rank retrieved articles according to important words in titles and abstracts; topics; author names; affiliations; journal names; publication year; and clustered by topic. Any result can be further mined by choosing any other button, and small search results can be expanded to include related articles. It has been deployed continuously, serving a wide range of biomedical users and needs, and over time has also served as a platform to support the creation of new tools that address additional needs. Here we describe the current, greatly expanded implementation of Anne O'Tate, which has added additional buttons to provide new functionalities: We now allow users to sort and rank search results by important phrases contained in titles and abstracts; the number of authors listed on the article; and pairs of topics that co-occur significantly more than chance. We also display articles according to NLM-indexed publication types, as well as according to 50 different publication types and study designs as predicted by a novel machine learning-based model. Furthermore, users can import search results into two new tools: e) Mine the Gap!, which identifies pairs of topics that are under-represented within set of the search results, and f) Citation Cloud, which for any given article, allows users to visualize the set of articles that cite it; that are cited by it; that are co-cited with it; and that are bibliographically coupled to it. We invite the scientific community to explore how Anne O'Tate can assist in analyzing biomedical literature, in a variety of use cases.

## Introduction

PubMed is the most used, and arguably most advanced search engine, for retrieving biomedical literature [https://www.ncbi.nlm.nih.gov/pubmed/, accessed May 15, 2020]. It has sophisticated automated features such as mapping queries to topics (Medical Subject Heading terms (MeSH)), imputing articles written by the same individual, spelling correction, and other tools

**Data Availability Statement:** All data are available from PubMed records (https://www.nlm.nih.gov/databases/download/pubmed_medline.html) and from iCite (https://icite.od.nih.gov/).

**Funding:** This research was supported by NIH Grants R01LM010817 and P01AG039347. The funders provided support in the form of salary for NS, but did not have any additional role in the study design, data collection and analysis, decision to publish, or preparation of the manuscript. The specific roles of authors are articulated in the 'author contributions' section.

that assist a wide variety of users who range from physicians and bench scientists to patients and their relatives. Despite this advanced user-focused engineering, the output of a PubMed query is a simple list of articles, and users have few options for summarizing or mining this output further.

We developed Anne O'Tate to be an integrated, generic tool for summarization, drill-down and browsing of PubMed search results that accommodates a wide range of biomedical users and needs [1]. Briefly, Anne O'Tate allows the user to carry out a PubMed query. After displaying the list of retrieved articles, the user can choose to visualize multiple aspects of the articles to the user, according to pre-defined categories such as the "most important" words found in titles or abstracts; topics; journals; authors; publication years; and affiliations. Clicking on a given item opens a new tab that displays all papers that contain that item. One can navigate by drilling down through the categories progressively, e.g., one can first restrict the articles according to author name and then restrict that subset by affiliation. Alternatively, one can expand small sets of articles to display the articles that are most closely related to the set as a whole.

Over the past two decades, about 20 web-based tools have been created to assist in biomedical retrieval and mining (reviewed in [2–5]), many of which have ceased to be maintained or to be offered as free services. Some of these tools, e.g., PubReMiner [6], allow users to sort articles by journal, author name, topic, etc., as does Anne O'Tate, and some, like Chilibot [7], offer specialized functionalities such as specific relationships between proteins, genes, or keywords.

However, Anne O'Tate remains unique both in supporting progressive drill-down, and in its range of advanced mining options. To date, the original Anne O'Tate paper [1] has been cited 57 times according to Google Scholar (excluding self-citations) and 16 times in PubMed Central. In 2017, Engwall reviewed Anne O'Tate from the standpoint of users, especially librarians [8]. He critiqued its slow performance for large queries (which we have since fixed by moving to a different server and adding memory) and its bare-bones interface (which we deliberately prefer as a design choice), but concluded that "Anne O'Tate provides an excellent tool set for searching PubMed. Its data mining tools provide a variety of dynamic content analysis that can be of great use in identifying relevant search terms and bibliometrics." [8].

As new user needs have become apparent, and as our own text-mining research programs have progressed, we have added a number of new facets to Anne O'Tate for summarizing and drilling-down the results of queries. We have also used Anne O'Tate as a platform to allow users to link PubMed queries to new, more specialized and advanced text processing programs. In this report, we describe the modifications and new functionalities that have been made since the initial report [1], and give use cases to demonstrate its broad usefulness to the biomedical community.

## Results

Suppose we enter the query [Alzheimer AND treatment] into the Anne O'Tate query interface (Fig 1). The query box is simplified compared to the PubMed home page but retains hotlinks so that the user can specify query Limits, and see and edit the exact query Details as processed by PubMed. The query is passed to PubMed, which processes the query and returns a list of the most recent 25,000 articles in reverse chronological order (Fig 2), querying PubMed [https://www.ncbi.nlm.nih.gov/pubmed/, accessed November 17, 2020]. Each user session is given a unique job ID and is saved in the webserver for approximately six months; the user can return to the most recent processed query by entering the Job ID into a separate query box on the homepage (Fig 1).

# Anne O'Tate

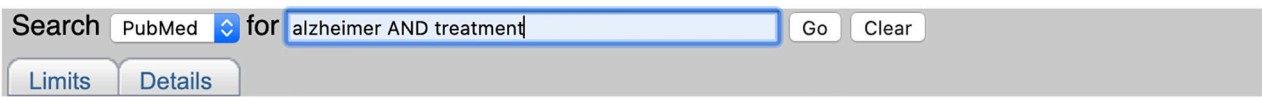

- Enter one or more search terms. Use PubMed help for more information.
- Enter author names as smith jc. Initials are optional.
- Enter journal titles in full or as MEDLINE abbreviations. Use the Journals Database to find journal titles.

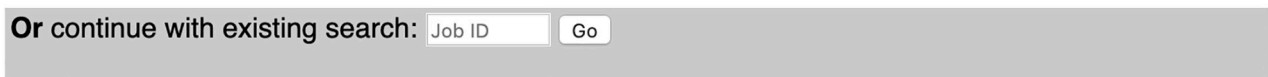

This search tool will help you gain an overview of the set of articles (up to 25,000 most recent articles) retrieved by a PubMed query. Once you enter a query, you can select different types of summary information to view.

Reference: Smalheiser NR, Zhou W, Torvik, VI. Anne O'Tate: A tool to support user-driven summarization, drill-down and browsing of PubMed search results. J Biomed Discov Collab. 2008; 3:2.

Last Modified: 05/15/2020

**Fig 1. Screenshot of the Anne O'Tate homepage.** Each displayed article (Fig 2) has two hotlinks on its right side: "Related articles" opens a new tab that displays a ranked list of the most related articles as computed by PubMed using the PubMed related article algorithm which is largely based on word usage similarity [9]. The "Citations" hotlink opens a new tab that displays the Citation Cloud surrounding that article (see below).

## Buttons for focused summarization, drill-down and analysis

On the left side of the page displaying the list of articles, there are 12 hotlinked facets or "buttons" that the user can choose to mine the set of retrieved articles further (Fig 2). Those buttons which are **new**** or have been **modified*** since the previous report on Anne O'Tate are marked with asterisks respectively in the following paragraphs. Each button opens up a new tab that displays a processed result. This can be viewed or processed further. We will discuss each button in turn:

**Important words***.   This computes the words that are significantly over-represented in the retrieval set (here, 42,500 articles on Alzheimer AND treatment) compared to all articles contained in the MEDLINE database. (Note that MEDLINE contains over 26 million publications curated from high-quality journals [10], representing the majority of articles covered by the PubMed search engine.) They are ranked in order of their "importance", i.e., the degree to which the word is over-represented [1, 11]. The list can be further filtered using a button to restrict the terms to one or more semantic categories (taken from the UMLS [12] as described in [1]). Clicking on any one initiates a new query restricted to articles that mention that word in any field of the article's PubMed metadata (title, abstract, or other metadata field). The important words are displayed in stemmed form [13]–for example, the listed word "amyloid"

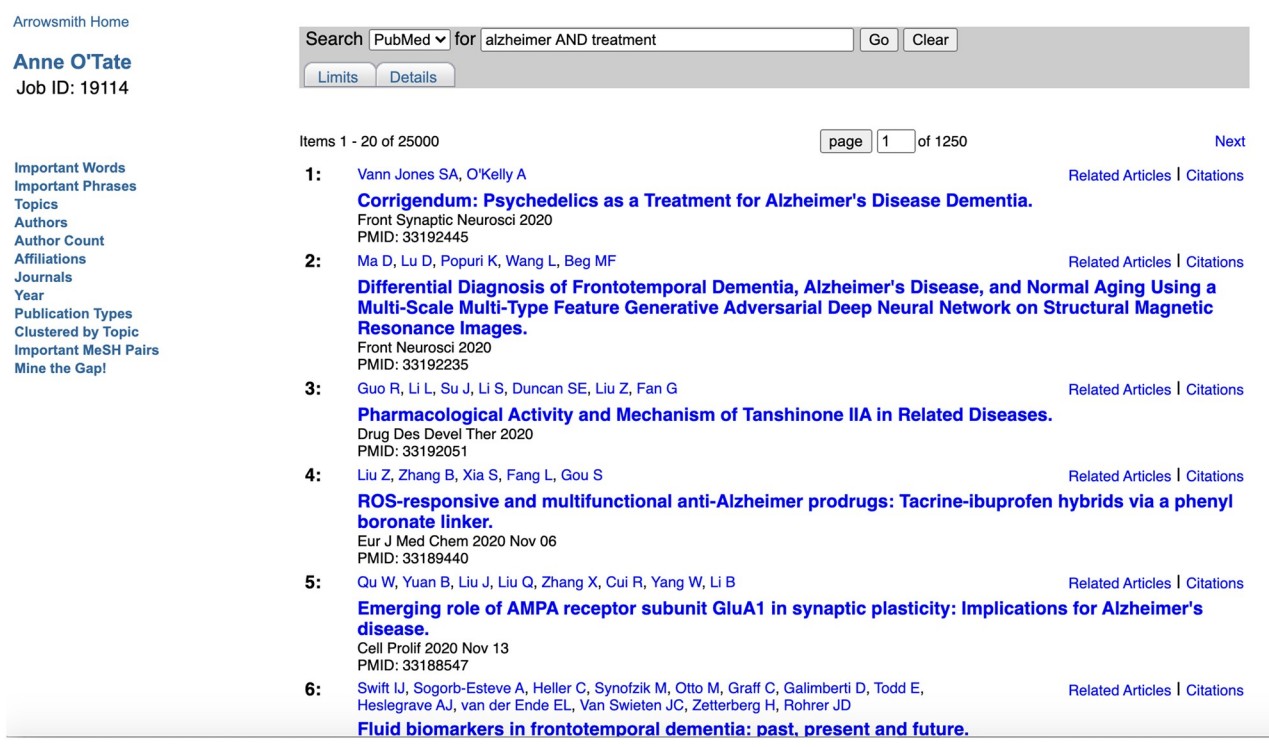

**Fig 2. Screenshot of the list of articles retrieved from PubMed using the query [Alzheimer AND treatment].**

in Fig 3 comprises mentions of both "amyloid" and "amyloids". Accordingly, clicking on the hotlinked term "amyloid" initiates a new, more restricted PubMed query: [Alzheimer AND treatment AND (amyloid [all fields] OR amyloids [all fields])], and the resulting list of 13,858 articles is displayed in a new tab (Fig 4). The list of Important Words together with their Importance scores [1, 11] can also be exported as a CSV file for use in text mining.

**Important phrases**\*\*. This runs the TopMine algorithm [14] to identify phrases that are important within the titles and abstracts of retrieved articles (without comparison to their frequency in MEDLINE). For the [Alzheimer AND treatment] query, we see phrases such as "Alzheimer disease", "cognitive function", and "oxidative stress" (Fig 5). Clicking on any phrase initiates a new restricted query [Alzheimer AND treatment AND "exact phrase" [tiab]] whose results are shown in a new tab. Note that, unlike Important Words, Important Phrases are not stemmed and they are mined only from titles and abstracts, not all fields of the PubMed record. The list of Important Phrases can also be exported as a CSV file for use in text mining.

**Topics.** This displays the Medical Subject Headings (MeSH terms) [15] indexed in the set of retrieved articles. Because the MeSH terms are listed in order of document frequency, and because the top twenty terms are found on the first page, the most prevalent terms are immediately visible to the user, though one can scroll to later pages to view less prevalent terms if desired.

**Authors.** This lists author names (defined as lastname, firstinitial) ranked by order of document frequency within the set of retrieved articles. This allows the user to rapidly filter the set of articles for those having a given name as co-author. No attempt is made here to disambiguate different individuals sharing the same name. However, we have disambiguated each author name on each PubMed article in a separate project, Author-ity [16, 17]. The Author-ity 2009 release is linked to Anne O'Tate: Each author name on each article in the displayed list is

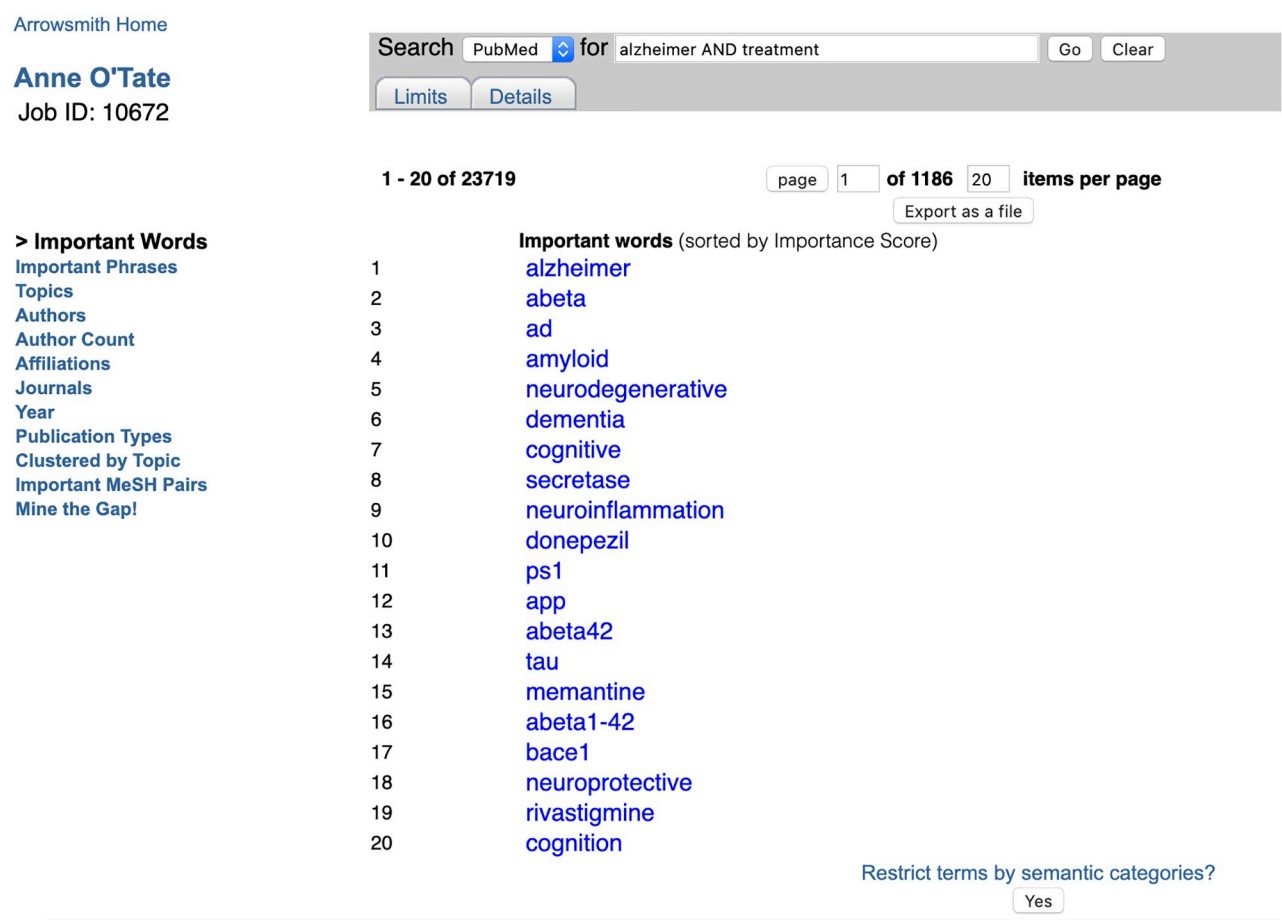

**Fig 3. Screenshot of important words calculated for the query [Alzheimer AND treatment].**

hotlinked, and clicking on it opens a new tab which displays the predicted individual if that article is present in the Author-ity 2009 dataset. The most recent Author-ity beta release contains articles through 2019, and is available upon request, but has not yet been disseminated in final form publicly nor linked to Anne O'Tate.

**Author count**\*\*. This button allows the user to choose articles that have a specified number or range of authors. One simply cannot query PubMed by author count, so this is a uniquely valuable feature that allows one to choose, for example, only those articles that are sole-authored, or those having more than five co-authors. Since prominent, senior investigators are the ones most likely to write reviews and editorials as sole-authored papers, a possible use case would be to identify likely senior scientists in a given field by carrying out a PubMed query on a given topic, then filtering first for sole-author articles, then for publication types Editorial and Review, and finally listing their author names ranked by frequency. Note that the Author Count simply counts the number of authors on each single paper, NOT the author names. If a paper is written by "John Smith and John Smith", the author count on that paper is 2.

**Affiliations.** This button displays chunks of text that are delimited by commas or periods in the Affiliation field of the PubMed record. For example, "University of Illinois" or "Cold Spring Harbor Laboratory" or "USA" are typical chunks that correspond to institutions, cities, or countries. Of all of the buttons employed in Anne O'Tate, this feature seems to confuse

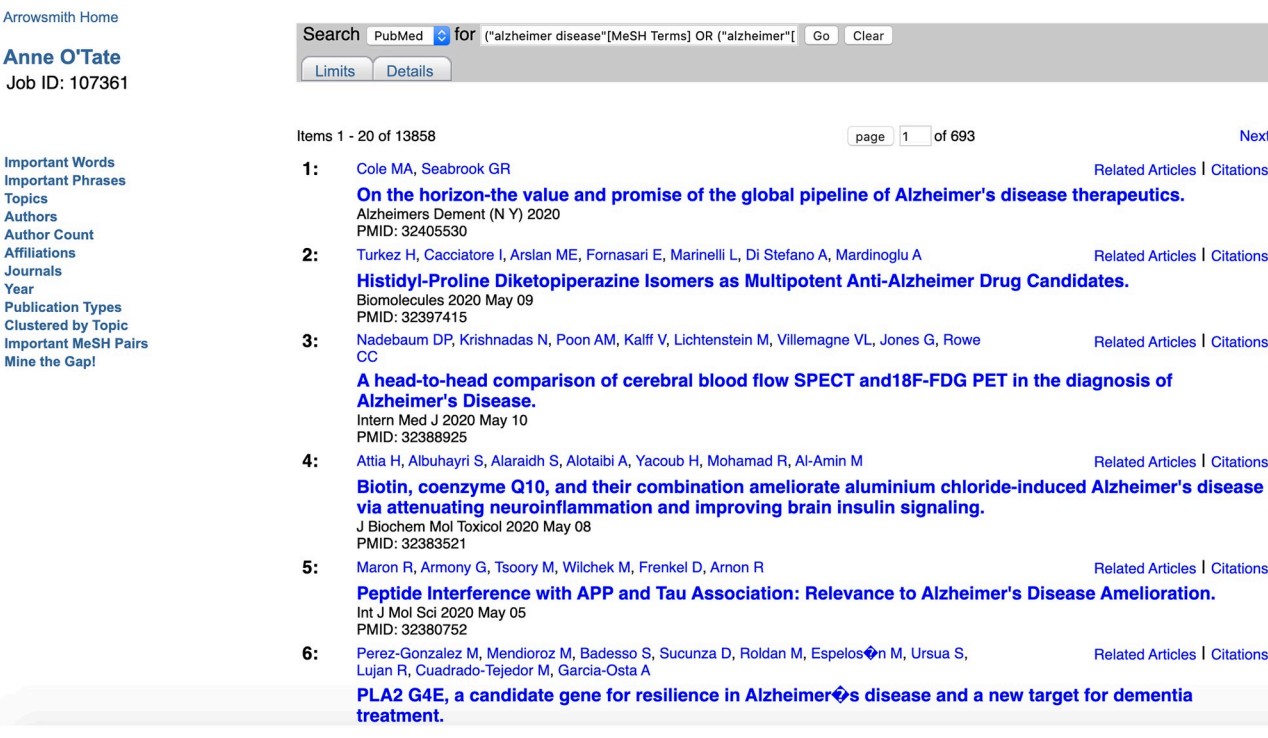

**Fig 4. Screenshot of the list of articles that mention "amyloid" or "amyloids" within the original query.**

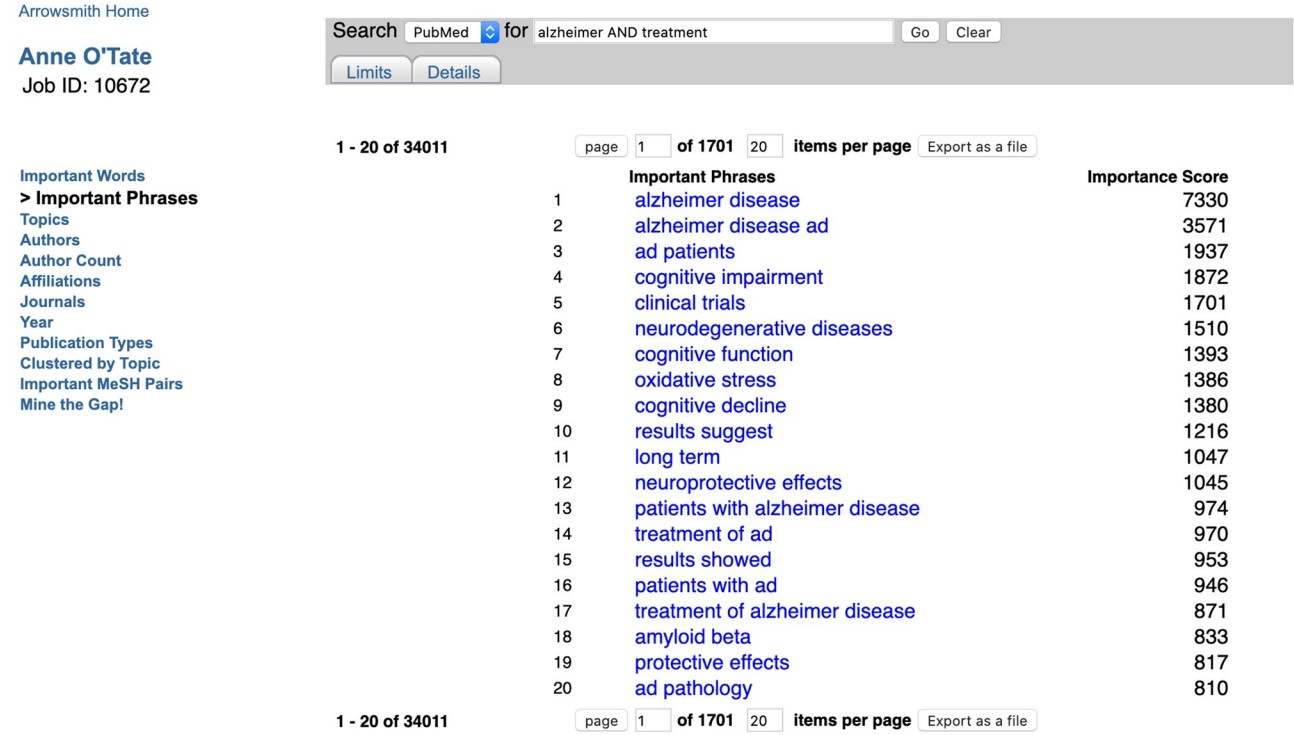

**Fig 5. Screenshot of important phrases calculated for the query [Alzheimer AND treatment].**

users the most [8], since the chunks are displayed in order of document frequency, rather than in a hierarchy (e.g., institution–city–state–country). Because of the simple text processing that is carried out, different chunks may be semantically redundant (e.g., NY and New York are treated separately), and an entity followed by a comma is displayed separately than the same entity followed by a period. In short, the Affiliations chunks give a somewhat "Cubist" representation of the nonstandard ways that different articles may fill out the Affiliation field. Nevertheless, instead of employing high-powered Natural Language Processing to reorganize the display output, we have decided not to revise this button, because our colleague Vetle Torvik has already developed an elegant tool, MapAffil, that maps articles rather precisely to their geographical locations [18, 19].

**Journals.** This displays the names of journals in the set of retrieved articles ranked by document frequency.

**Year.** This displays the distribution of publication dates by year for the set of retrieved articles. The list can be displayed as a histogram and downloaded if desired.

**Publication types**\*\*. This displays the distribution of publication types as indexed by MEDLINE or assigned by PubMed for the set of retrieved articles. The publication types are grouped into categories: **Problematic** publication types include Retracted Publication, Published Erratum, Retraction of Publication, and Corrected and Republished Article. **Clinical Trials** comprises 10 different types of articles that include the word "trial" (e.g., Pragmatic Clinical Trial) as well as Multicenter Study. **Clinical Studies** extends the list of Publication Types to include certain clinically important study designs (that are indexed by MeSH terms), such as cohort studies, case-control studies, cross-over studies, prospective and retrospective studies, etc. **Article Types** list all other articles (e.g., review, case report, practice guideline, etc.) in descending order of document frequency. Finally, **Research Support** lists sources of research support (e.g., NIH, Extramural or Non-U.S. Gov't) in descending order of frequency.

For the publication types and study designs shown in Table 1, we have implemented a probabilistic machine-learning based model that, given an article, will predict the probability that the article satisfies the criteria to be indexed as one of the multiple publication types and clinically important study designs. If an article is predicted to belong to one of these Publication Types but is not displayed in the left column, it is displayed on the right under "Additional Predicted" articles (Fig 6). We also display articles that our model predicts are concerned with Diagnostic Test Accuracy (which currently lacks an official NLM indexing tag).

The model will be described in detail in upcoming publications (Aaron Cohen et al, in preparation). Briefly, positive training sets were collected for each of 50 publication types or study designs by taking a random sample of PubMed articles that were indexed with that category (publication type [Publication Type] or study design [MeSH term: no expansion]). We represented each article as a multi-dimensional vector of features taken from title, abstract, and metadata; among the features were unigrams and bigrams and several text similarity representations (paragraph2vec [20] and our own implicit text similarity representation [21]). Then, each publication type or study design was summarized as a single vector by averaging the vectors of the individual articles in that category. The vector distances of a given PubMed article to each of the category vectors was used to predict the probability p that it belongs to one or more of the categories. To implement the model in the Publication Types button in Anne O'Tate, we set a different threshold value of p for each category (i.e., set at the threshold having a precision of 0.8 on a test set; if no threshold exhibits that precision, then it is set at the threshold having the highest F1 performance) and display articles under "Additional Predicted" if their predictive score p is over threshold but they lack NLM indexing for that category.

**Table 1. Publication types and study designs for augmented display and prediction.**

| |
| --- |
| AUTOBIOGRAPHY |
| BIBLIOGRAPHY |
| BIOGRAPHY |
| CASE CONTROL STUDY |
| CASE REPORTS |
| CLINICAL STUDIES AS TOPIC |
| CLINICAL STUDY |
| COHORT STUDY |
| COMMENT |
| CONGRESSES |
| CONSENSUS DEVELOPMENT CONFERENCE |
| CROSS CULTURAL COMPARISON |
| CROSS OVER STUDY |
| CROSS SECTIONAL STUDY |
| DIAGNOSTIC TEST ACCURACY |
| DOUBLE BLIND METHOD |
| EDITORIAL |
| EVALUATION STUDY |
| EVALUATION STUDIES AS TOPIC |
| FEASIBILITY STUDIES |
| FOCUS GROUPS |
| FOLLOW UP STUDIES |
| GENOME WIDE ASSOCIATION STUDY |
| HISTORICAL ARTICLE |
| HUMAN EXPERIMENTATION |
| INTERVIEW |
| INTERVIEWS AS TOPIC |
| LECTURES |
| LEGAL CASES |
| LETTER |
| LONGITUDINAL STUDIES |
| MATCHED PAIR ANALYSIS |
| META ANALYSIS |
| MULTICENTER STUDY |
| NEWS |
| PERSONAL NARRATIVES |
| PORTRAITS |
| PRACTICE GUIDELINE |
| PREDICTIVE VALUE OF TESTS |
| PROSPECTIVE STUDIES |
| RANDOM ALLOCATION |
| RANDOMIZED CONTROLLED TRIAL |
| REPRODUCIBILITY OF RESULTS |
| RETROSPECTIVE STUDIES |
| REVIEW |
| SYSTEMATIC REVIEW |
| SYSTEMATIC REVIEW AS TOPIC |
| TWIN STUDY |
| VALIDATION STUDY |

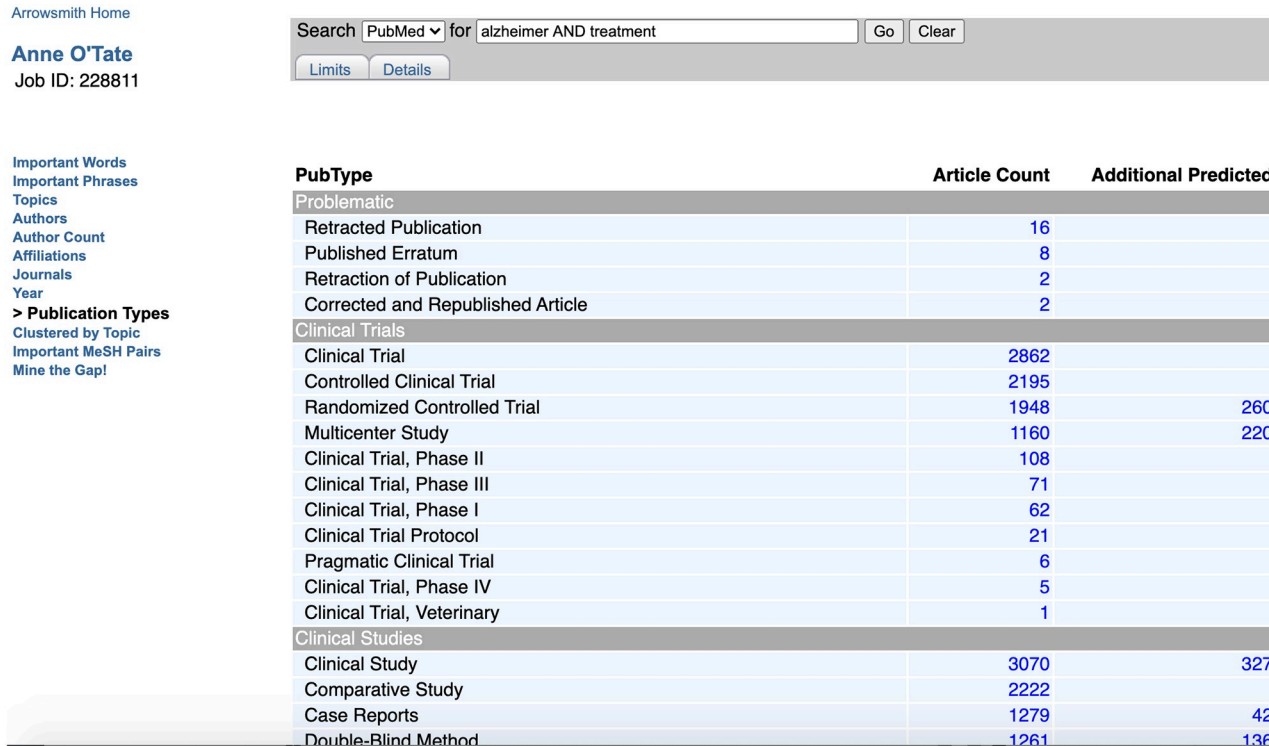

**Fig 6. Screenshot showing a sample query with both indexed and predicted Randomized Controlled trial articles displayed.** The query [Alzheimer AND treatment] was further processed using the Publication Types button, and the initial part of the display including Problematic Publications and Clinical Trials is shown. Note that 1948 articles are indexed under Randomized Controlled Trials [PT], whereas another 260 articles are predicted to be RCTs as well according to our model.

In this fashion, we will supplement the current indexing of PubMed, allowing us to a) classify the most recently published articles which have not been indexed yet, b) articles which are not included in MEDLINE and so lack PT and MH indexing, and c) articles which failed to be indexed by MEDLINE indexers, perhaps because of their constraints (e.g., if an article can be indexed at more than one level of the MeSH hierarchy, manual indexers only apply the most specific MeSH term, whereas our model would classify the article as belonging to more than one level).

**Clustered by topic.** This runs an algorithm which partitions the set of retrieved articles into 12 different topical categories, as evenly as possible in terms of the number of articles in each category [1]. This can be useful in surveying the range of topics that are covered in the set, without being unduly influenced by the most common or the most rare.

**Important MeSH pairs**[**]**.** This considers pairs of MeSH terms that co-occur on individual articles within the set of retrieved articles. For those pairs which co-occur on at least four articles within the set, it displays and ranks those pairs according to the odds ratio (i.e., the frequency within the set divided by frequency within MEDLINE as a whole). Important MeSH pairs often represent either frank relations (e.g., "Melatonin" is secreted by the "Pineal Gland"), semantically related MeSH terms ("Prion Diseases" vs. "Prions"), or implicit relationships (e.g., "Polysomnography" is used to diagnose "Sleep Apnea, Obstructive"). This may help to characterize the kinds of relations studied within a set of retrieved articles, and to drill down more precisely than if one were choosing individual topics.

**Mine the gap!**[**]**.** This button runs a tool that identifies "gaps" within the set of retrieved articles, i.e., pairs of MeSH terms that do NOT co-occur on any article within the set, even

though the individual MeSH terms occur in more than 10% of the set, such that $> 10$ articles would be predicted to co-occur just by chance [22]. Finding gaps in a literature may help to predict which new lines of research may be undertaken in the future. Generally, one only finds gaps satisfying those criteria in sets that contain at least several hundred articles. Clicking on the Mine the Gap! button starts a program that identifies and displays gaps, along with some of their features [22]. An optional button carries out Arrowsmith two-node searches [23, 24] on each gap. This allows the user to view and analyze terms that may bridge the gap [24], and calculates the pR ratio for each gap—this is a measure of the amount of implicit information shared by the pair of MeSH terms [23], and (all other factors equal) gaps that share a lot of implicit information are more likely to be worth studying further. Because the tool and its underlying model are rather specialized and have been described elsewhere in detail, the reader is directed to its original publication for use cases [22].

### Drill down and expansion of queries

An important architectural feature of Anne O'Tate is that any new tab displaying a list of articles can be further processed according to any of the 12 buttons on the left side, facilitating easy, progressive, multi-faceted "drill-down" and "slice-and-dice" of the retrieval set as desired. For example, if one would like to find authors who have written recent review articles, one can do the topical query, then click on the Publication Types button, then choose the Review button, then click on the Year button, then choose a given year, and then click on the Authors button to display the list of author names whose articles satisfy all the chosen criteria.

Conversely, any list of displayed articles which is smaller than 50 can be expanded to add new articles that are highly related to multiple articles in the original set. Clicking on the "expand" button at the top of the page runs an algorithm that employs the PubMed related articles algorithm [9] in batch mode [1] and adds related articles to the original set.

### The citation cloud**

This tool provides a service that is not achieved directly through PubMed or any other public service, and should greatly enable the study of citations by the scientific community. Clicking on "Citations" next to any displayed article opens a new tab that allows the user to visualize the "citation cloud" around that article: That is, the set of articles cited by it; those which cite it; those which are co-cited with it; and those which are bibliographically coupled to it (Fig 7).

To say that article A and B are co-cited means that A and B are both cited by one or more articles $C_i$ [25]. Co-citation is a measure of similarity that is not based on textual or topical similarity. Note that the co-citation relationship is not fixed but can vary over time depending on how many newer articles cite both A and B. In contrast, bibliographically coupled (BC) articles cite some of the same articles in their reference lists [26]. Bibliographic coupling provides another measure of similarity that is not directly based on textual or topical similarity, and has the distinct advantage that the BC relationship can be calculated for any two articles regardless of when they are published. As well, the BC relationship is stable and will not change over time. Citation, co-citation, and bibliographic coupling networks have been studied extensively, alone or combined with text-based similarity measures, for classifying articles and for large-scale mapping of science fields [27].

Clicking on any box (Fig 7) opens a new tab that displays the list of articles in the box. The Citation Cloud tool employs a large dataset of open citations, including iCite [28] as well as other sources. The tool, and some of its typical use cases, are described in detail in a separate publication [29].

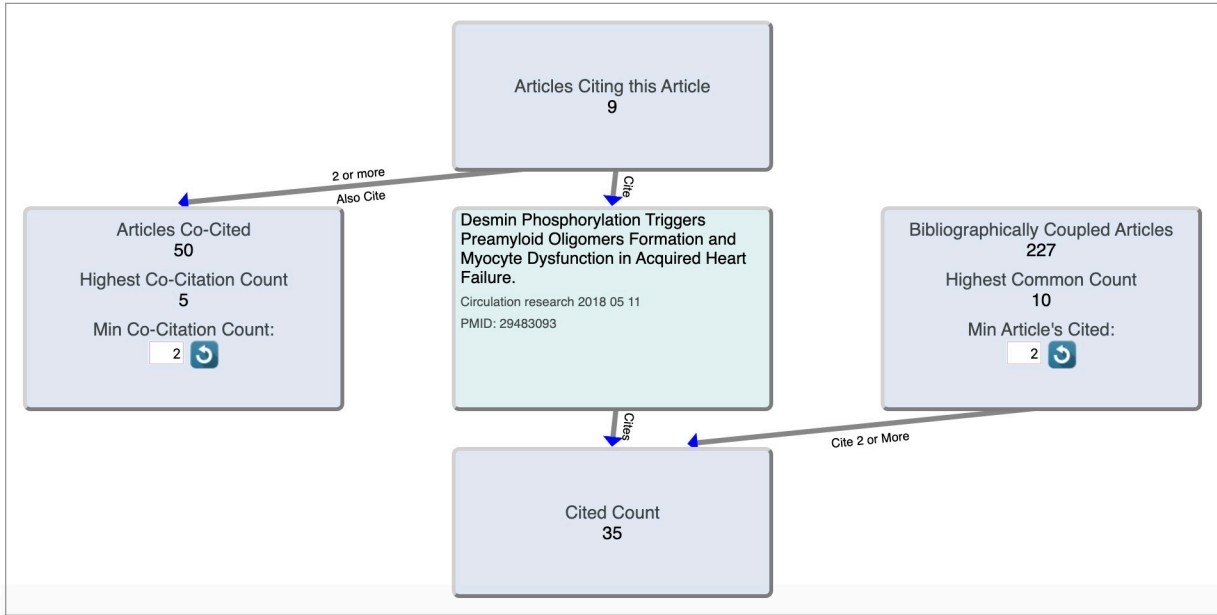

**Fig 7. Screenshot of the citation cloud for the target article: Rainer et al., desmin phosphorylation triggers preamyloid oligomers formation and myocyte dysfunction in acquired heart failure.** Circ Res. 2018 May 11;122(10):e75-e83.

## Discussion

A plethora of tools have been developed to assist in searching the biomedical literature (reviewed in [2–5]), though few have been maintained as a free, public service continuously to the present. Although some of these tools do provide faceted summarization (e.g., listing author and journal names for a set of retrieved article) [6], no other tool offers the unique architecture of Anne O'Tate to drill-down progressively according to any of a dozen dimensions, or to expand up again. Nor do any other search web servers serve as open platforms for adding new tools that greatly enhance the ability of users to analyze literature in a sophisticated manner, e.g., Mine the Gap! which identifies research gaps in the set of retrieved articles [22], and the Citation Cloud which displays the entire local network of citations surrounding any given article [29]. We can offer tools that may be too specialized for the government to host, and can add or change functionalities in a more nimble fashion.

Use cases for Anne O'Tate arise on a daily basis. For example, to find suggested reviewers for a given manuscript or proposal, one can enter a topical query (e.g., keywords taken from the title of the manuscript) and examine the list of authors who have published PubMed articles on that topic, especially those who have authored review articles. Similarly, to help decide which journal is best for submitting your own manuscript, one can enter a short description as a query and examine the list of journals which have published similar work. The Important Words and Important Phrases buttons may help to suggest new keywords for refining queries during high recall retrieval projects, such as accumulating evidence for a systematic review. Using the Publication Types button allows a user to track articles in a given set that have been

retracted, and the Citation Cloud to trace articles that have subsequently cited them anyway [30, 31]. Using the Year button, one can immediately see the trend of growth over time of a given topic. Using the Important MeSH Pairs, one can identify relationships that are specifically studied in the set of retrieved articles. Conversely, to identify relationships that have surprisingly NOT been studied at all [22], one can use the Mine the Gap! button.

We envision Anne O'Tate as a value-added layer on top of the PubMed search engine, but it worth emphasizing that many of its features are unique and allow users to perform queries that simply cannot be carried out in PubMed: a) For example, those studying biomedical innovation and discovery [e.g., 32] may wish to retrieve just those articles which have been authored by a single individual, or which have been authored by a large team, which can be readily found using our Author Count button but which is not possible to specify in PubMed. b) Those seeking to find articles that are topically relevant to a given article (e.g., in a high recall literature search as may be conducted in preparing a systematic review, in carrying out due diligence, or in properly citing related research when writing a scientific article) may not be satisfied with examining the PubMed related article list. They may also wish to retrieve the articles that are co-cited and bibliographically coupled to the given article. Again, this is not possible in PubMed but is readily found using our Citation Cloud button. c) The manual indexing of PubMed articles is limited by the fact that articles may not receive indexing for several months after publication, and that certain metadata such as publication types and MeSH terms are only assigned to the subset of articles indexed in MEDLINE. The Publication Types button classifies the list of retrieved articles according to 50 predicted publication types and study designs that goes a long way to overcoming these limitations. Other analytical features that are unique to Anne include Important Words, Important Phrases, clustered by topic, important MeSH pairs, and "Mine the Gap!".

It should be noted that a few months ago, PubMed changed its user interface and back end processing so that it is designed primarily for biomedical users of limited searching experience rather than librarians and data-literate searchers. In contrast, Anne O'Tate was designed to align with the previous version of PubMed. Entering a search into the new PubMed query box will give the same displayed results as a search into the Anne O'Tate query box, when visualizing the retrieved articles in reverse chronological order, though newly added PubMed options (e.g., showing Best Match) may cause a discrepancy in how the articles are ranked between the two tools.

We expect that Anne O'Tate will continue to evolve over time. For example, the new updated Author-ity author name disambiguation dataset (through 2019) has not yet been officially released; when it has, we expect to link it to the articles retrieved through Anne O'Tate so that any author name on any article can be disambiguated. As well, note that Anne O'Tate currently only processes the most recent 25,000 articles from any query (Fig 1). In part, this is to avoid slow-downs that may be caused from users entering huge, poorly formed, or malicious queries, but another reason is that some downstream functions involve sending lists of PMIDs to PubMed and there is a limit to the number of PMIDs that can be entered as a query. If power users of Anne O'Tate require larger queries, we are willing to work with them to increase these limits. Future changes will be driven by our own ongoing research, and by suggestions and feedback from the biomedical and informatics communities.

## Materials and methods

Many of the methods were discussed in detail in the original publications [1, 22, 29]. As one of the reviewers of Anne O'Tate pointed out [8], in the past Anne O'Tate tended to be slow in processing large queries. We therefore devoted considerable effort to optimizing the data

retrieval techniques used for passing information between entities Anne O'Tate and PubMed. PubMed only allows a certain number of articles to be fetched in a single request, even when using an API key (to validate us as a known user) and functions optimally on numbers less than that maximum.

PubMed also only allows a certain number of PMIDs (as a list) to be sent within a single query dictated also by maximum HTTP request size. Internally within our server, the SQL database that holds an internal parsed representation of all PubMed records also has a practical limit within queries. To overcome these limits, batch processing was used when necessary, operating within each component's limits within each batch yet still operating on the complete data set in whole, so that queries were processed smoothly by all components of the system. On two of the new buttons–- Year and Publication Types—the user-facing client-side uses AJAX requests which allow for the long running processes to be more elegantly and efficiently returned to the end user. The end user in these cases sees immediate results to the screen in real time, as the data are pulled from PubMed and/or collected locally. The previous version used timed browser refreshes, which check with the server at each refresh to see if the server-side function has completed. These refreshes sometimes appeared to 'hang' when the server-side processes died inadvertently, leaving the client to refresh infinitely. The errors causing server-side processes to die were fixed, resolving the 'hanging' problem.

Some other changes made to the original implementation of Anne O'Tate were the elimination of the system-level kill commands being issued on the background Perl daemon processes, and more granular batch processing of the data pipeline (as mentioned above). These made user jobs more stable, faster and less likely to hang and resulted in fewer server crashes. The Architecture supporting Anne O'Tate is comprised of LINUX Ubuntu Server 18.04 LTS, Perl 5 version 26, Python 2.7 and 3.6 and MySql 5.7.

## Acknowledgments

We thank the users of Anne O'Tate who have given feedback over the years, and who have emailed us whenever the site went down or a bug was detected! A preprint of a draft of this paper was previously deposited in the University of Illinois at Chicago Indigo Data Repository [33].

## Author Contributions

**Conceptualization:** Neil R. Smalheiser.

**Formal analysis:** Dean P. Fragnito, Eric E. Tirk.

**Funding acquisition:** Neil R. Smalheiser.

**Investigation:** Neil R. Smalheiser.

**Methodology:** Neil R. Smalheiser, Dean P. Fragnito, Eric E. Tirk.

**Project administration:** Neil R. Smalheiser.

**Software:** Dean P. Fragnito, Eric E. Tirk.

**Supervision:** Neil R. Smalheiser.

**Writing – original draft:** Neil R. Smalheiser.

**Writing – review & editing:** Neil R. Smalheiser, Dean P. Fragnito, Eric E. Tirk.

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
