## [Decision Letter · Decision Letter 0]

13 Aug 2020

PONE-D-20-15023

Anne O’Tate: Value-added PubMed search engine for analysis and text mining

PLOS ONE

Dear Dr. Smalheiser,

Thank you for submitting your manuscript to PLOS ONE. After careful consideration, we feel that it has merit but does not fully meet PLOS ONE’s publication criteria as it currently stands. Therefore, we invite you to submit a revised version of the manuscript that addresses the points raised during the review process.

We look forward to receiving your revised manuscript.

Kind regards,

Rezarta Islamaj, PhD

Academic Editor

PLOS ONE

Additional Editor Comments:

Dear author,

the reviewers have raised some reasonable concerns regarding the manuscript. Please take them into account. We look forward to your next communication

Journal Requirements:

We note that one or more of the authors are employed by a commercial company: Xornet Inc.

2.1. Please provide an amended Funding Statement declaring this commercial affiliation, as well as a statement regarding the Role of Funders in your study. If the funding organization did not play a role in the study design, data collection and analysis, decision to publish, or preparation of the manuscript and only provided financial support in the form of authors' salaries and/or research materials, please review your statements relating to the author contributions, and ensure you have specifically and accurately indicated the role(s) that these authors had in your study. You can update author roles in the Author Contributions section of the online submission form.

2.2. Please also provide an updated Competing Interests Statement declaring this commercial affiliation along with any other relevant declarations relating to employment, consultancy, patents, products in development, or marketed products, etc. 

Reviewers' comments:

Reviewer's Responses to Questions

**Comments to the Author**

1. Is the manuscript technically sound, and do the data support the conclusions?

Reviewer #1: Yes

Reviewer #2: Partly

Reviewer #3: Partly

Reviewer #4: Partly

2. Has the statistical analysis been performed appropriately and rigorously? 

Reviewer #1: N/A

Reviewer #2: N/A

Reviewer #3: N/A

Reviewer #4: No

3. Have the authors made all data underlying the findings in their manuscript fully available?

Reviewer #1: Yes

Reviewer #2: No

Reviewer #3: Yes

Reviewer #4: No

4. Is the manuscript presented in an intelligible fashion and written in standard English?

Reviewer #1: Yes

Reviewer #2: Yes

Reviewer #3: Yes

Reviewer #4: Yes

5. Review Comments to the Author

Reviewer #1: The authors present the recent functional updates of Anne O’Tate, a web tool that enhances PubMed search engine to provide users with several additional functions to analyze the search results. This manuscript if published will help publicize the tool and serve as a tutorial for users. I only have a couple minor comments:

1. Adding an introduction to MEDLINE and MeSH terms will help readers who are not familiar with these terms to understand better on the methodologies and output of the tool.

2. Is the ranking of retrieved results taken into consideration (weighted differently) in summarizing on Topics, Authors, Affiliations etc.?

Reviewer #2: The authors present a nice overview of their Anne O’Tate tool for querying the PubMed/Medline database. All functions were described with details, as well as the methods behind them. However, it is still not clear which functions are actually new and I miss good examples (use cases) for many of the functions proposed in the tool.

Major review:

- The authors implemented many functions in the tool, but I miss a real use case which could demonstrate that some (interesting) insights could only have been learned by using some of the proposed function and not with Pubmed itself or other available tool.

- Indeed, I miss short discussion of other available tool, since many tools are indeed available, and how these compared to Anne O’Tate.

Minor review:

- The abstract does not mention what is new in the tool in comparison to its older versions.

- The introduction does not cite how many visits (e.g., per day, or in total) the tool has had in the last years, nor whether it has been helpful to support the research from others (e.g., publications that cited Anne O'Tate). I also miss a short overview of what are the new features that will be describe in this new publication.

- I am not sure whether some of functions are indeed helpful. For instance, by using raw frequency, the most popular surnames will always be on the top of the author list. It was also not clear to me whether the author count could be indeed helpful. I miss examples or use cases on how these functions might help the search.

- Further, for the affiliation function, it is indeed confusing, since it is a mix of countries, cities, departments and universities. Further, “USA” appeared twice: “USA” and “USA.”

- I found that the example of an earthquake in Puerto Rico was not suitable (off topic) since the manuscript so far had been showing the use case of treatment for Alzheimer disease.

- “Mine the Gap!” is indeed a creative function, but top results were pairs that probably do not make sense together, e.g. “Caregivers::Mice, Transgenic”, even though some might be interesting, e.g., “Mice, Transgenic::Surveys and Questionnaires”. Again, a good use case showing its utility would help.

Reviewer #3: This study presents the recent updates of Anne O’Tate, a web server provides additional analytics to PubMed query results. It introduces primary features such as important words, important phrases, topic clustering and many others. Altogether it provides rich information over the retrieved pmids. Please find my comments below.

I evaluated the system using the query “covid-19 diabetes” and found a few issues:

(1) The relevance of top k pmids is of concern. The top 5 retrieved PMIDs were 32634827, 32634717, 32634716, 32634459, 32633728. They do have the term ‘diabetes’, but diabetes is not the main topic, so I am not sure why they are ranked at the top. Comparatively, the top 5 pmids from both PubMed and LitCovid are much more relevant – the content is directly on diabetes. The paper claims that it passes the query to PubMed, but the retrieved results are significantly different. And this is not because of publication dates (the study uses the PubMed data up to May) as some of the top retrieved pmids from PubMed were published before May. The retrieved pmids are critical since all the features are based on them. I suspect there are some issues for query expansion and translation, before passing to PubMed.

(2) The results of ‘Affiliations’ for this query is confusing. The top affiliations are a mixed of countries (e.g., China), cities (e.g., Milan), universities (e.g., Huazhong University of Science and Technology), and departments (e.g., Department of Medicine). So what exactly is an affiliation? Huazhong University of Science and Technology is a university in China. Why showing them separately? It would be much better to represent in a hierarchical manner. Post-processing the Affiliation field of PubMed records is essential. The current results are confusing and not informative.

(3) The results of ‘Clustered by topic’ also have problems. When clicking the “Most recent articles” cluster, it shows an error message “Request-URI Too Long”. Also, the topic “AMP-Activated Protein Kinases” has only 1 pmid, why this is considered as a topic?

(4) For the results of ‘important words’, somehow ‘PubMed’ is listed as an important word.

In addition, while it introduces many features, there is no evaluation detailed in the paper. Some of the adopted methods were published a while ago; also the study cited a few methods that are in preparation without giving details. The effectiveness of these methods is unclear. The study should provide a substantial evaluation of the methods and compares with existing methods as baselines.

Minor comments:

(1) Please provide an API which supports batch processing.

(2) What is the usage of the system, and how often does the system update?

(3) For line 207-209, the font is different.

Reviewer #4: This paper is a brief introduction to Anne O'Tate, which adds additional functionalities to PubMed and intends to help PubMed users.

Although some interesting gadgets are proposed and implemented, this paper fails to show its effectiveness. PLOS One audience would like to read about how the new features would help them, both qualitatively and quantitatively compared with PubMed. Some features are truly unavailable on PubMed or other literature search engine, the authors did not quantify how the features can create results which are not available elsewhere.

This paper could have been more persuasive with more evidences and statistics.

Specifically I'd like to see the following additional information:

1. Experiments on important words/phrases: from Anne O'tate web log, find some examples and show how Important scores make sense in determining importance of words/phrases.

2. Due to no disambiguation on author names, how much of Author count is accurate.

3. How effective 'Mine the Gap!' can find gaps and predict future research lines.

4. In Citation Cloud, bibliographically coupled articles may include some cases which are not based on similarity, how often such cases happened and how to minimize their damage to the results.

6. PLOS authors have the option to publish the peer review history of their article (what does this mean?). If published, this will include your full peer review and any attached files.

Reviewer #1: No

Reviewer #2: No

Reviewer #3: No

Reviewer #4: No

---

## [Author Response · Author response to Decision Letter 0]

30 Sep 2020

we have uploaded the response to reviewers as a separate document, but will cut and paste it here too. 

Response to Reviewers 

Reviewer #1: The authors present the recent functional updates of Anne O’Tate, a web tool that enhances PubMed search engine to provide users with several additional functions to analyze the search results. This manuscript if published will help publicize the tool and serve as a tutorial for users. I only have a couple minor comments:

1. Adding an introduction to MEDLINE and MeSH terms will help readers who are not familiar with these terms to understand better on the methodologies and output of the tool.

 We have now added more information about what MeSH terms are and what MEDLINE is, including a reference. 

2. Is the ranking of retrieved results taken into consideration (weighted differently) in summarizing on Topics, Authors, Affiliations etc.?

 We have stated that in the initial query results, articles are displayed in reverse chronological order, not by predicted relevance. We also state for each button how articles are ranked – sometimes by frequency, sometimes by importance scores which have been defined in our previous publication. 

Reviewer #2: The authors present a nice overview of their Anne O’Tate tool for querying the PubMed/Medline database. All functions were described with details, as well as the methods behind them. However, it is still not clear which functions are actually new and I miss good examples (use cases) for many of the functions proposed in the tool.

We have now added use cases to the paper and discussed them in some detail. Also, we have now clarified in Results that we have added one asterisk to indicate buttons that have been modified substantially since our previous publication, and two asterisks to indicate entirely new buttons. 

Major review:

- The authors implemented many functions in the tool, but I miss a real use case which could demonstrate that some (interesting) insights could only have been learned by using some of the proposed function and not with Pubmed itself or other available tool.

Done. 

- Indeed, I miss short discussion of other available tool, since many tools are indeed available, and how these compared to Anne O’Tate.

In the Discussion we cite several reviews that have discussed similar tools in detail, including a comparison to Anne O’tate. 

Minor review:

- The abstract does not mention what is new in the tool in comparison to its older versions.

We have now extensively rewritten the abstract to include the original set of buttons and to emphasize specific new buttons that have been added. 

- The introduction does not cite how many visits (e.g., per day, or in total) the tool has had in the last years, nor whether it has been helpful to support the research from others (e.g., publications that cited Anne O'Tate). I also miss a short overview of what are the new features that will be describe in this new publication.

We have now added text in the Introduction pointing out several groups who have reviewed Anne’s performance (before the recent improvements) and compared it to other existing tools. The added text to the abstract should provide the requested overview of new features up front. 

- I am not sure whether some of functions are indeed helpful. For instance, by using raw frequency, the most popular surnames will always be on the top of the author list. It was also not clear to me whether the author count could be indeed helpful. I miss examples or use cases on how these functions might help the search.

We have now added a use case for the Author Count function, and added more description of the Author Names and how to use them. We are now displaying Authors by lastname firstinitial rather than simply by lastname, which will help with disambiguation somewhat. We also alert the reader that we have disambiguated authors on papers in a separate project which is being released publicly through 2019 (even if the most recent papers are not yet linked directly to Anne yet). 

- Further, for the affiliation function, it is indeed confusing, since it is a mix of countries, cities, departments and universities. Further, “USA” appeared twice: “USA” and “USA.”

We have now added further text to describe how to use the Affiliation chunks. 

- I found that the example of an earthquake in Puerto Rico was not suitable (off topic) since the manuscript so far had been showing the use case of treatment for Alzheimer disease.

We have now changed the example to be consistent. 

- “Mine the Gap!” is indeed a creative function, but top results were pairs that probably do not make sense together, e.g. “Caregivers::Mice, Transgenic”, even though some might be interesting, e.g., “Mice, Transgenic::Surveys and Questionnaires”. Again, a good use case showing its utility would help.

We have now added more description and a use case here (although this tool is fully described in detail in a separate publication that is now a few years old).

Reviewer #3: This study presents the recent updates of Anne O’Tate, a web server provides additional analytics to PubMed query results. It introduces primary features such as important words, important phrases, topic clustering and many others. Altogether it provides rich information over the retrieved pmids. Please find my comments below.

I evaluated the system using the query “covid-19 diabetes” and found a few issues:

(1) The relevance of top k pmids is of concern. The top 5 retrieved PMIDs were 32634827, 32634717, 32634716, 32634459, 32633728. They do have the term ‘diabetes’, but diabetes is not the main topic, so I am not sure why they are ranked at the top. Comparatively, the top 5 pmids from both PubMed and LitCovid are much more relevant – the content is directly on diabetes. The paper claims that it passes the query to PubMed, but the retrieved results are significantly different. And this is not because of publication dates (the study uses the PubMed data up to May) as some of the top retrieved pmids from PubMed were published before May. The retrieved pmids are critical since all the features are based on them. I suspect there are some issues for query expansion and translation, before passing to PubMed.

 As mentioned above and stated in the paper, the retrieved articles are listed in reverse chronological order, NOT in predicted relevance. But beyond that, it is important to note that PubMed changed their public interface in a very major way this spring, rather suddenly, so that their new interface works very differently than it had for many years. We built our tool to be compatible with the old PubMed interface, and our tool is not altered at all – however, it does not always coincide with the way that PubMed displays results, depending on user settings. We have now added text to the Discussion to alert users to the possible discrepancy – which does not affect the utility of our tool in any way. 

(2) The results of ‘Affiliations’ for this query is confusing. The top affiliations are a mixed of countries (e.g., China), cities (e.g., Milan), universities (e.g., Huazhong University of Science and Technology), and departments (e.g., Department of Medicine). So what exactly is an affiliation? Huazhong University of Science and Technology is a university in China. Why showing them separately? It would be much better to represent in a hierarchical manner. Post-processing the Affiliation field of PubMed records is essential. The current results are confusing and not informative.

 As mentioned above, we have now clarified that the method of displaying Affiliations by chunks may be initially confusing and give hints how to use the results. We also now cite the work of Torvik who has developed a geographical mapping tool for PubMed articles, which may be used as an alternative to our Affiliations button. 

(3) The results of ‘Clustered by topic’ also have problems. When clicking the “Most recent articles” cluster, it shows an error message “Request-URI Too Long”. Also, the topic “AMP-Activated Protein Kinases” has only 1 pmid, why this is considered as a topic?

 The reviewer chose a very unusual query – covid-19 is an extremely new topic so almost all of the papers lack MeSH terms which are employed for clustering. The set of papers actually clustered is thus quite small, and we always divide the set into 12 partitions, which explains why AMP-activated protein kinases was included. Normally, the bulk of retrieved articles are not extremely new, so the bulk of articles are clustered. As for the URI-error, this is a limitation of entering PMIDs into the PubMed query box, which we discussed in the Methods section, and we have now addressed the issue by fetching the records another way. 

(4) For the results of ‘important words’, somehow ‘PubMed’ is listed as an important word.

 This is not a bug. PubMed IS mentioned in many covid-19 articles, e.g., systematic reviews where they say that they searched PubMed for relevant articles. 

In addition, while it introduces many features, there is no evaluation detailed in the paper. Some of the adopted methods were published a while ago; also the study cited a few methods that are in preparation without giving details. The effectiveness of these methods is unclear. The study should provide a substantial evaluation of the methods and compares with existing methods as baselines.

 We have now given several additional use cases which should be persuasive that the tool has value for biomedical users. The algorithmic aspects are described and evaluated in detail in previous publications, and the newest ones (e.g. predicted publication types) will be described in detail in separate publications. We are confident that the results that we give are “correct” and transparent. However, the value of Anne O’Tate is not so much the uniqueness of the algorithms but the fact that they are packaged together into one seamless integrated platform. There is no comparable tool. 

Minor comments:

(1) Please provide an API which supports batch processing.

 The reviewer is asking for us to build an extensive option which is not part of the system, and which no one has ever asked us to build before. I don’t think that we can do this as a condition of revising the paper, although if there is serious user interest we would consider it. 

(2) What is the usage of the system, and how often does the system update?

 We have not kept track of recent usage since we moved the web server to a new server machine and expanded memory. In any case, we do not expect that user load will be a significant factor in performance, even if the user base expands after this paper is published. The Anne “system” is a set of web-based tools and does not have a formal release (unlike software). The updates to the system are described in this paper, so you could think of this paper as describing Anne 2.0. 

(3) For line 207-209, the font is different.

 Fixed. 

Reviewer #4: This paper is a brief introduction to Anne O'Tate, which adds additional functionalities to PubMed and intends to help PubMed users.

Although some interesting gadgets are proposed and implemented, this paper fails to show its effectiveness. PLOS One audience would like to read about how the new features would help them, both qualitatively and quantitatively compared with PubMed. Some features are truly unavailable on PubMed or other literature search engine, the authors did not quantify how the features can create results which are not available elsewhere.

This paper could have been more persuasive with more evidences and statistics.

Specifically I'd like to see the following additional information:

1. Experiments on important words/phrases: from Anne O'tate web log, find some examples and show how Important scores make sense in determining importance of words/phrases.

 We have previously described Important Words in two papers and it is not one of the new functionalities emphasized in the current paper (although we have updated its display to a minor extent). Important words and phrases is shown in figures 4 and 5 for a specific search (Alzheimer and treatment) where the sense of the displayed words and phrases is hopefully self-evident. 

2. Due to no disambiguation on author names, how much of Author count is accurate.

 We have now clarified that the Author Count simply counts the number of authors on each single paper, NOT the author names. If a paper is written by “John Smith and John Smith”, the author count on that paper is 2. 

3. How effective 'Mine the Gap!' can find gaps and predict future research lines.

 This is cited in our previous publication and discussed there. 

4. In Citation Cloud, bibliographically coupled articles may include some cases which are not based on similarity, how often such cases happened and how to minimize their damage to the results.

 We have now clarified that bibliographically coupled (BC) articles are related in respect to partially shared reference lists , and very well may not be textually similar. This is actually a good thing, since BC and text similarity are two complementary types of similarity. We have now added citations to people who have discussed and studied this before.

---

## [Decision Letter · Decision Letter 1]

3 Nov 2020

PONE-D-20-15023R1

Anne O’Tate: Value-added PubMed search engine for analysis and text mining

PLOS ONE

Dear Dr. Smalheiser,

Thank you for submitting your manuscript to PLOS ONE. After careful consideration, we feel that it has merit but does not fully meet PLOS ONE’s publication criteria as it currently stands. Therefore, we invite you to submit a revised version of the manuscript that addresses the points raised during the review process.

We look forward to receiving your revised manuscript.

Kind regards,

Rezarta Islamaj, PhD

Academic Editor

PLOS ONE

Additional Editor Comments (if provided):

Dear Author,

Thank you for the revisions provided in response to the reviewers' suggestions. Two of the reviewers feel that some technical details are still needed in order to properly substantiate the conclusions. Please revise the manuscript to your best ability, and I will review again,

Thank you

Reviewers' comments:

Reviewer's Responses to Questions

**Comments to the Author**

1. If the authors have adequately addressed your comments raised in a previous round of review and you feel that this manuscript is now acceptable for publication, you may indicate that here to bypass the “Comments to the Author” section, enter your conflict of interest statement in the “Confidential to Editor” section, and submit your "Accept" recommendation.

Reviewer #1: All comments have been addressed

Reviewer #2: (No Response)

Reviewer #3: All comments have been addressed

Reviewer #4: (No Response)

2. Is the manuscript technically sound, and do the data support the conclusions?

Reviewer #1: Yes

Reviewer #2: Yes

Reviewer #3: Yes

Reviewer #4: Partly

3. Has the statistical analysis been performed appropriately and rigorously? 

Reviewer #1: Yes

Reviewer #2: N/A

Reviewer #3: N/A

Reviewer #4: N/A

4. Have the authors made all data underlying the findings in their manuscript fully available?

Reviewer #1: Yes

Reviewer #2: No

Reviewer #3: Yes

Reviewer #4: Yes

5. Is the manuscript presented in an intelligible fashion and written in standard English?

Reviewer #1: Yes

Reviewer #2: Yes

Reviewer #3: Yes

Reviewer #4: Yes

6. Review Comments to the Author

Reviewer #1: The authors have addressed my concerns from the initial review. I recommend acceptance of the manuscript.

Reviewer #2: One of the items in the major review was not addressed properly. The authors did not provide a proper use case in the discussion, but just hypotetical examples of how the tool could potentially be used in this or that situation. I meant a proper biomedical use case, e.g., looking for a treatment for disease X, and the various steps of how the searching was done in the tool and new knowledge has been found, etc. And that when doing such search in a similar way in PubMed, proving that no similar results could be obtained.

Reviewer #3: My comments have been addressed properly. Thanks to the authors for the efforts. Having an API would be useful for potential users to retrieve results systematically.

Reviewer #4: This version provides numerous additional information. A user case is added to help understand how to use this web site and interpret the results. An impressive addition to the publication types certainly increases the value of the service.

The following are some issues with the draft:

1. The draft mentioned machine-learning based model to predict publication type. Please provide some details about the model.

2. As pointed by other reviewers, when affiliation button was tested, most results showed similar mixture of country, city, institution, department names. A user of this button expects to see a list of entities at the same level, for example, a list of countries or a list of universities or institutions. For the query ‘Alzheimer AND treatment’, affiliation button retrieves more than 1, 000 pages of results. Without a search function, it is hard for the users to find the one they are interested in. Current design confused users and it needs more work to be useful.

3. Topic button simply uses MeSH terms to represent topic classification. Although a unified topic system is convenient and consistent, it is very small granularity. As a result, the same Alzheimer query returns more than 490 pages of results. Again, without a search function, the results are simply overwhelming for users.

4. Figure 2 displays some unicode characters incorrectly, although the website shows them correctly. Please update this figure.

7. PLOS authors have the option to publish the peer review history of their article (what does this mean?). If published, this will include your full peer review and any attached files.

Reviewer #1: No

Reviewer #2: No

Reviewer #3: No

Reviewer #4: No

---

## [Author Response · Author response to Decision Letter 1]

18 Nov 2020

Reviewer #2: One of the items in the major review was not addressed properly. The authors did not provide a proper use case in the discussion, but just hypotetical examples of how the tool could potentially be used in this or that situation. I meant a proper biomedical use case, e.g., looking for a treatment for disease X, and the various steps of how the searching was done in the tool and new knowledge has been found, etc. And that when doing such search in a similar way in PubMed, proving that no similar results could be obtained.

 Thank you for this comment. In the revised version, we now added text to the Discussion to indicate more clearly three practical use cases that would be easy in Anne but impossible in PubMed. Anne O’Tate is designed to be extremely broad in adding value for a wide range of use cases; many of our features do not exist in PubMed and could not be conducted via PubMed searches. For example, our progressive drill-down architecture, our expand button, Important Words, Important Phrases, author count, predicted publication types, clustered by topic, important MeSH pairs, Mine the Gap! and Citation Cloud are all unique to Anne. We already indicated how these would be used and why, in the Results section. 

Reviewer #4: This version provides numerous additional information. A user case is added to help understand how to use this web site and interpret the results. An impressive addition to the publication types certainly increases the value of the service.

The following are some issues with the draft:

1. The draft mentioned machine-learning based model to predict publication type. Please provide some details about the model.

 In the revised version, we have now added the following description of the model: Briefly, positive training sets were collected for each of 50 publication types or study designs by taking a random sample of PubMed articles that were indexed with that category (publication type [Publication Type] or study design [MeSH term: no expansion]). We represented each article as a multi-dimensional vector of features taken from title, abstract, and metadata; among the features were unigrams and bigrams and several text similarity representations (paragraph2vec and our own implicit text similarity representation). Then, each publication type or study design was summarized as a single vector by averaging the vectors of the individual articles in that category. The vector distances of a given PubMed article to each of the category vectors was used to predict the probability p that it belongs to one or more of the categories. To implement the model in the Publication Types button in Anne O’Tate, we set a different threshold value of p for each category (having the highest F1 performance on a test set) and display articles under “Additional Predicted” if their predictive score p is over threshold but they lack NLM indexing for that category. 

2. As pointed by other reviewers, when affiliation button was tested, most results showed similar mixture of country, city, institution, department names. A user of this button expects to see a list of entities at the same level, for example, a list of countries or a list of universities or institutions. For the query ‘Alzheimer AND treatment’, affiliation button retrieves more than 1, 000 pages of results. Without a search function, it is hard for the users to find the one they are interested in. Current design confused users and it needs more work to be useful.

 We understand that the reviewer would like to see an entirely different architecture for the Affiliations button. However, this button is not newly described in the present paper. Rather, it has been implemented for 12 or so years and was described in our previous paper. We have neither had any user complaints nor have encountered use cases where we needed to re-engineer this button. We will take the reviewer’s advice to heart for future evolution of the system, but we do not feel that changing the Affiliations button should be a precondition for publishing the present paper.

3. Topic button simply uses MeSH terms to represent topic classification. Although a unified topic system is convenient and consistent, it is very small granularity. As a result, the same Alzheimer query returns more than 490 pages of results. Again, without a search function, the results are simply overwhelming for users.

 In the revised version, we have now clarified that because the MeSH terms are listed in order of document frequency, and because the top twenty terms are found on the first page, the most prevalent terms are immediately visible to the user, though one can scroll to later pages to view the less prevalent terms if desired. Most of the 490 pages represents a long tail of MeSH terms found in only one article, hence are not very important at all. 

4. Figure 2 displays some unicode characters incorrectly, although the website shows them correctly. Please update this figure.

 The reviewer has sharp eyes! We now made a new figure 2 that does not show incorrect Unicode characters. (By the way, the Unicode mapping is a rather subtle issue. The retrieval of articles from PubMed does handle Unicode characters, but a few of the downstream functions involve mapping author names to normalized ascii formats (i.e., in the Authors button and in links out to the Author-ity tool) and so ascii is still displayed in a few places deep within Anne O’Tate. Users can manually replace the ascii characters with Unicode but that is not necessary for proper functioning.)

---

## [Decision Letter · Decision Letter 2]

13 Jan 2021

PONE-D-20-15023R2

Anne O’Tate: Value-added PubMed search engine for analysis and text mining

PLOS ONE

Dear Dr. Smalheiser,

Thank you for submitting your manuscript to PLOS ONE. After careful consideration, we feel that it has merit but does not fully meet PLOS ONE’s publication criteria as it currently stands. Therefore, we invite you to submit a revised version of the manuscript that addresses the points raised during the review process.

We look forward to receiving your revised manuscript.

Kind regards,

Rezarta Islamaj, PhD

Academic Editor

PLOS ONE

Additional Editor Comments (if provided):

Dear authors,

It seems important to address the issue of the use case raised by Reviewer 2, and comment on the additional feature suggested by Reviewer 4. Please take these suggestions into consideration, thanks

Reviewers' comments:

Reviewer's Responses to Questions

**Comments to the Author**

1. If the authors have adequately addressed your comments raised in a previous round of review and you feel that this manuscript is now acceptable for publication, you may indicate that here to bypass the “Comments to the Author” section, enter your conflict of interest statement in the “Confidential to Editor” section, and submit your "Accept" recommendation.

Reviewer #1: All comments have been addressed

Reviewer #2: (No Response)

Reviewer #3: All comments have been addressed

Reviewer #4: All comments have been addressed

2. Is the manuscript technically sound, and do the data support the conclusions?

Reviewer #1: Yes

Reviewer #2: Yes

Reviewer #3: Yes

Reviewer #4: Yes

3. Has the statistical analysis been performed appropriately and rigorously? 

Reviewer #1: Yes

Reviewer #2: N/A

Reviewer #3: N/A

Reviewer #4: N/A

4. Have the authors made all data underlying the findings in their manuscript fully available?

Reviewer #1: Yes

Reviewer #2: No

Reviewer #3: Yes

Reviewer #4: Yes

5. Is the manuscript presented in an intelligible fashion and written in standard English?

Reviewer #1: Yes

Reviewer #2: Yes

Reviewer #3: Yes

Reviewer #4: Yes

6. Review Comments to the Author

Reviewer #1: The authors have addressed my concerns. I don’t have further concerns/comments about the revised manuscript.

Reviewer #2: Regarding my request for a proper use case, it hasn't been addressed by the authors. They simply listed three hypothetical, high-level situations, but they did not design a proper use case. The tool has been available since more than 10 years, but they could not prove, by citing a reference, that it has been used by other research groups to support solving any particular task. This was a request from me that the authors did not address. Therefore, I think that propely describing a use case which (somehow) prove the utility of the the tool is crucial to be sure that this new version is really necessary and that the task could not be solved using other existing tools.

Reviewer #3: My primary comments have been addressed. Thanks for the dedicated efforts. I do not have further comments.

Reviewer #4: Most issues have been addressed or explained.

I will make a small recommendation to the topic classification functionality. Currently the articles from different topics are displayed without the number of articles, while other buttons (Journals, pub types) display number of articles. It's good information for the user to know when they search.

I understand it may take time to implement but I really like to have this feature as a user.

7. PLOS authors have the option to publish the peer review history of their article (what does this mean?). If published, this will include your full peer review and any attached files.

Reviewer #1: No

Reviewer #2: No

Reviewer #3: No

Reviewer #4: No

---

## [Author Response · Author response to Decision Letter 2]

22 Jan 2021

Dear Dr. Islamaj,

Thank you for allowing me to revise the paper a third time. I hope that you will be able to make a decision without returning it for further reviews. I have made a good faith effort to satisfy the concerns of both reviewers. 

Reviewer #4: Most issues have been addressed or explained.

I will make a small recommendation to the topic classification functionality. Currently the articles from different topics are displayed without the number of articles, while other buttons (Journals, pub types) display number of articles. It's good information for the user to know when they search.

I understand it may take time to implement but I really like to have this feature as a user.

 Thank you for your suggestion. We have now implemented this functionality as requested by the reviewer. 

Reviewer #2: Regarding my request for a proper use case, it hasn't been addressed by the authors. They simply listed three hypothetical, high-level situations, but they did not design a proper use case. The tool has been available since more than 10 years, but they could not prove, by citing a reference, that it has been used by other research groups to support solving any particular task. This was a request from me that the authors did not address. Therefore, I think that properly describing a use case which (somehow) prove the utility of the tool is crucial to be sure that this new version is really necessary and that the task could not be solved using other existing tools.

 As much as we would like to satisfy the reviewer, I think there is some misunderstandings that need to be clarified. 

The second reviewer feels, in this and the previous review, that we should prove that Anne O’Tate can find biomedical information which cannot be found directly through PubMed searches alone (e.g. new treatments for a given disease) -- and that other biomedical researchers have used and cited Anne as having found such unique information. However, the original Anne O’Tate paper never made such claims, nor is that the intent of the original tool. The original tool was aimed at summarization, drill-down and mining of sets of articles retrieved from PubMed. It is NOT a literature-based discovery tool like Arrowsmith. Therefore, the type of documentation that the reviewer is asking for is not appropriate. 

In the new version of Anne O’Tate that we present in the submitted manuscript, we do indeed introduce several NEW kinds of functionalities, that do find information that cannot be retrieved directly through PubMed searches (for example, author count, predicted publication types, and co-citations and bibliographically coupled articles in the citation cloud). Each of these is described and documented with specific examples. However, since these are brand-new and just now being reported, one cannot expect that these new functions have yet been used or cited by other research groups. Also, since these new functions are retrieving bibliographic rather than biomedical information, they are not likely to produce new knowledge such as new suggested treatments. (Though millions of people use search engines such as PubMed or Google Scholar to assist them, I do not believe that anyone has cited the use of these search engines in their research papers.)

In the final version of this ms., we DID cite references to show that others have reviewed and inspected Anne O’Tate, both from the technical as well as user points of view, and found it to have value for people carrying out literature searches: 

“However, Anne O’Tate remains unique both in supporting progressive drill-down, and in its range of advanced mining options. To date, the original Anne O’Tate paper [1] has been cited 57 times according to Google Scholar (excluding self-citations) and 16 times in PubMed Central. In 2017, Engwall reviewed Anne O’Tate from the standpoint of users, especially librarians [8]. He critiqued its slow performance for large queries (which we have since fixed by moving to a different server and adding memory) and its bare-bones interface (which we deliberately prefer as a design choice), but concluded that “Anne O’Tate provides an excellent tool set for searching PubMed. Its data mining tools provide a variety of dynamic content analysis that can be of great use in identifying relevant search terms and bibliometrics.” [8]. 

I hope that this good-faith effort will suffice for publishing this manuscript.

---

## [Editor Report · Decision Letter 3]

25 Feb 2021

Anne O’Tate: Value-added PubMed search engine for analysis and text mining

PONE-D-20-15023R3

Dear Dr. Smalheiser,

We’re pleased to inform you that your manuscript has been judged scientifically suitable for publication and will be formally accepted for publication once it meets all outstanding technical requirements.

Kind regards,

Rezarta Islamaj, PhD

Academic Editor

PLOS ONE
---

## [Editor Report · Acceptance letter]

26 Feb 2021

PONE-D-20-15023R3 

Anne O’Tate: Value-added PubMed search engine for analysis and text mining 

Dear Dr. Smalheiser:

I'm pleased to inform you that your manuscript has been deemed suitable for publication in PLOS ONE. Congratulations! Your manuscript is now with our production department. 

Kind regards, 

on behalf of

Dr. Rezarta Islamaj 

Academic Editor

PLOS ONE